# Plasma trimethylamine N-oxide and its metabolic precursors and risk of mortality, cardiovascular and renal disease in individuals with type 2-diabetes and albuminuria

**Signe Abitz Winther** [1,2]\*, **Jens Christian Øllgaard**[2], **Tine Willum Hansen**[1], **Bernt Johan von Scholten**[2], **Henrik Reinhard**[1], **Tarunveer Singh Ahluwalia**[1], **Zeneng Wang**[3], **Peter Gæde**[4,5], **Hans-Henrik Parving**[6], **Stanley Hazen**[3], **Oluf Pedersen**[7], **Peter Rossing**[1,8]

1 Steno Diabetes Center, Copenhagen, Denmark, 2 Novo Nordisk A/S, Bagsvaerd, Denmark, 3 Department of Cardiovascular and Metabolic Sciences, Lerner Research Institute, Cleveland Clinic, Cleveland, OH, United States of America, 4 Slagelse Hospital, Slagelse, Denmark, 5 Univeristy of Southern Denmark, Odense, Denmark, 6 Rigshospitalet, Copenhagen, Denmark, 7 Novo Nordisk Foundation Center for Basic Metabolic Research, Copenhagen, Denmark, 8 Univeristy of Copenhagen, Copenhagen, Denmark

\* signe.abitz.winther@regionh.dk

**Data Availability Statement:** As data contains potentially identifying patient information, access to

## Abstract

### Aims

The trimethylamine N-oxide (TMAO) pathway is related to intestinal microbiota and has been associated to risk of cardiovascular disease (CVD). We investigated associations between four plasma metabolites in the TMAO pathway and risk of all-cause mortality, CVD and deterioration in renal function in individuals with type 2-diabetes (T2D) and albuminuria.

### Materials and methods

Plasma concentrations of TMAO, choline, carnitine, and betaine were measured by liquid chromatography-tandem mass spectrometry at baseline in 311 individuals with T2D and albuminuria. Information on all-cause mortality and fatal/non-fatal CVD during follow-up was obtained from registries. The association of each metabolite, and a weighted sum score of all four metabolites, with the endpoints were examined. Serum creatinine was measured at follow-up visits and the renal endpoint was defined as eGFR-decline of $\geq 30\%$. Associations were analysed using proportional hazards models adjusted for traditional risk factors.

### Results

Baseline mean(SD) age was 57.2(8.2) years and 75% were males. Follow-up was up to 21.9 years (median (IQR) follow-up 6.8 (6.1–15.5) years for mortality and 6.5 (5.5–8.1) years for CVD events). The individual metabolites and the weighted sum score were not associated with all-cause mortality (n = 106) or CVD (n = 116) (adjusted p≥0.09). Higher choline, carnitine and the weighted sum score of the four metabolites were associated with

data requires approved protocol from the the Capital Region Committee on Health Research Ethics (vek-anmeldelse.center-for-regional-udvikling@regionh.dk) and approved data sharing agreement with the Capital Region (cru-fp-vfd@regionh.dk). Thereafter data can be provided from Steno Diabetes Center Copenhagen (stenoinfo.stenodiabetescentercopenhagen@regionh.dk, attention "Complications Research").

**Funding:** This work was funded by the Novo Nordisk Foundation, grant no. NNF 14OC0013659 as part of the PROTON (Personalizing Treatment of diabetic Nephropathy) study. The Novo Nordisk Foundation Center for Basic Metabolomics Research is an independent research center at the University of Copenhagen partially funded by an unrestricted donation from the Novo Nordisk Foundation (www.cbmr.ku.dk) (grant number NNF18CC0034900). SLH reports funding by NIH grant P01 HL147823 and the Leducq Foundation. SAW was funded by The Innovation Fund Denmark (grant number 5016-00150B) and Novo Nordisk A/S provided support in the form of salary during the PhD fellowship for author SAW. JO and BJvS are after data collection employed by Novo Nordisk A/S. The specific roles of these authors are articulated in the 'author contributions' section. The funding sources did not have any additional role in the study design, data collection and analysis, decision to publish, or preparation of the manuscript.

**Competing interests:** I have read the journal's policy and the authors of this manuscript have the following competing interests: PR reports having given lectures for Astra Zeneca, Bayer, Novo Nordisk and Boehringer Ingelheim, and has served as a consultant for AbbVie, Astra Zeneca, Bayer, Eli Lilly, Boehringer Ingelheim, Astellas, Gilead, Mundipharma, Vifor, and Novo Nordisk, all fees given to Steno Diabetes Center Copenhagen. SAW, TWH, BJvS, TSA and PR own stocks in Novo Nordisk A/S and TSA in Zealand Pharma A/S . Z.W. and S.L.H. are named as coinventors on pending and issued patents held by the Cleveland Clinic relating to cardiovascular diagnostics and therapeutics, and have the right to receive royalty payment for inventions or discoveries related to cardiovascular diagnostics or therapeutics from Cleveland Heart Lab, Quest Diagnostics and Proctor & Gamble. S.L.H. also reports having been paid as a consultant from Proctor & Gamble, and having received research funds from Proctor & Gamble and Roche. Novo Nordisk A/S provided support in the form of salaries for authors SAW, JO and BJvS. This does not alter our adherence to

higher risk of decline in eGFR (n = 106) (adjusted p = 0.001, p = 0.03 and p<0.001, respectively).

## Conclusions

In individuals with T2D and albuminuria, higher choline, carnitine and a weighted sum of four metabolites from the TMAO pathway were risk markers for deterioration in renal function during long-term follow-up. Metabolites from the TMAO pathway were not independently related to risk of all-cause mortality or CVD.

## Introduction

The prevalence of type 2-diabetes (T2D) is increasing world-wide, accompanied by a cumulative burden of micro- and macrovascular complications. Even though prevention and treatment of cardiovascular disease (CVD) and renal complications of T2D have improved, the incidence is still high and the outcome often remains lethal. Moreover, an improved survival with increasing incidence of diabetes is resulting in increasing prevalence of renal complications [1].

Metabolites produced by the intestinal microbiota through digestion of certain dietary products are suggested to be involved in the development of metabolic diseases and risk of CVD, progression of kidney disease and mortality [2, 3]. Recently, there has been particular focus on the trimethylamine N-oxide (TMAO) pathway [4, 5]. A fraction of bacteria in the gut digest choline and L-carnitine from diet sources such as red meat, eggs and cheese and produce trimethylamine (TMA). In the liver TMA is oxidized to TMAO by flavin monooxygenases, and thereafter circulated to various tissues or excreted through the kidneys.

Previous studies have linked circulating concentrations of TMAO and its metabolic precursors (choline, L-carnitine and betaine) to mortality and adverse outcomes of CVD including cohorts of subjects with T2D or pre-existing CVD [6–8]. Plasma TMAO concentrations have also been associated with development of renal insufficiency both in subjects with normal kidney function and in chronic kidney disease [3, 9]. TMAO modulates lipid metabolism and increases vascular cell activation and endogenous inflammation thereby promoting atherogenesis [2, 10, 11]. However, it has been suggested that the adverse effect of TMAO may be mediated by impaired renal function or other mediators [9, 12] and it is still discussed whether the precursors of TMAO are associated with risk of mortality, CVD and renal outcomes independently of TMAO [13–15]. Hence, the role of TMAO and its precursors as risk markers for mortality, CVD and renal outcomes need further clarification. Therefore, in the present follow-up study we investigated associations between TMAO and its metabolic precursors (choline, L-carnitine and betaine) and risk of mortality, CVD and renal function decline in individuals with T2D and microalbuminuria.

## Material and methods

### Study subjects

We combined two Danish cohorts originating from the outpatient clinic at Steno Diabetes Center Copenhagen. Both cohorts included individuals with T2D and albuminuria. Design, method and follow-up for the original studies have been described in detail previously [16–19]. Briefly, the first cohort included individuals from the Steno-2 trial (ClinicalTrials.gov

PLOS ONE policies on sharing data and materials. The other authors report no conflicts of interest.

registration no. NCT00320008). This was a randomized controlled trial of 160 T2D individuals with microalbuminuria (n = 138 with available metabolite measures for the present study) receiving conventional versus intensified multifactorial treatment [16]. Inclusion criteria included individuals with T2D (defined according to the 1985 WHO criteria), age 40–65 years and persistent microalbuminuria. Microalbuminuria was defined as albumin excretion rate (UAER) of 30–300 mg in four of six 24-h urine samples. The trial was initiated in 1993 with end of intervention after a mean of 7.8 years, and thereafter continued as an observational study. The second cohort comprised 200 T2D individuals with persistent albuminuria (two out of three consecutive measured UAER > 30 mg/24h). Other inclusion criteria included age between 20 and 70 years, no known coronary artery disease and normal plasma creatinine. Of the 200 participants included in the original observational study, 173 had available metabolite measures for the present study. These participants were examined in 2007–08 with the original purpose to investigate presence of asymptomatic coronary artery disease in T2D individuals with albuminuria [18]. All participants received intensive multifactorial treatment constituting of glycaemic, lipid and blood pressure control, as well as antithrombotic therapy and lifestyle modification. Individuals from the original two studies with available metabolite measures from stored blood samples were included in the present study. Fifteen individuals participated in both studies and are in the present analyses included with baseline measures of the metabolites and clinical variables from the participation in the Steno-2 trial. The present study complies with the Declaration of Helsinki and was approved by the local ethics committee (the Capital Region Committee on Health Research Ethics, approval number H-17004111). All individuals gave written informed content at the original recruitment.

## Measurements of bioclinical variables

Plasma concentrations of TMAO, choline, carnitine and betaine were measured in samples collected at baseline. All samples were centrifuged, immediately stored in freezers at -80˚C until analysis and subsequently they were analysed at the same time. Plasma concentrations of TMAO, choline, carnitine and betaine were measured using stable isotope dilution high-performance liquid-chromatography with online electrospray ionization-tandem mass spectrometry (LC/MS/MS) on an AB-SCIEX5000 triple quadrupole mass spectrometer using d9-(trimethyl)-labelled internal standards as described previously [20] and performed in the Lerner Research Institute, Cleveland Clinic, by investigators blinded to any participants' characteristics. Quantification of plasma concentrations of TMAO, choline, carnitine and betaine each had a limit of quantification of 20 nmol/L. Quality control samples were run with each batch, and the inter-day coefficient-of-variation for each analyte was less than 10% over a span of 20 days.

UAER was measured in 24-h urine collections by an enzyme immunoassay. Until September 1st 2004, serum creatinine levels were measured by a modified Jaffe method and thereafter by an isotope dilution-mass spectrometry method. Therefore, all creatinine measurements determined by Jaffe were corrected to standardize measurements [21]. Estimated glomerular filtration rate (eGFR) was calculated using the Chronic Kidney Disease Epidemiology Collaboration (CKD-EPI) equation [22]. Serum cholesterol concentrations were determined by standard methods. HbA1c was measured by high-performance liquid-chromatography and current smoking was defined as one or more cigarettes/cigars/pipes a day.

## Endpoints

Through the Danish nationwide registers, all participants were tracked by their unique and personal identification number to obtain information on all hospital admissions and cause of

death. Reporting to these registries is mandatory and has high validity [23]. One participant was lost to follow-up after 13 years due to emigration. Extracted diagnoses for the two cohorts have previously been specified [17, 19]. We considered four endpoints; All-cause mortality, CV mortality, CVD and eGFR decline ≥30%. All deaths were classified as CVD related unless an unequivocal non-CV cause was reported, a previously recognized approach [24]. CVD events were defined as a composite of cardiovascular mortality, non-fatal myocardial infarction, ischaemic CVD, non-fatal stroke, amputation due to ischemia and cardiac or peripheral revascularization. Individuals experiencing multiple events stopped their follow-up at time of the first event in the analyses.

The renal endpoint was defined as time to the first occurrence of ≥30% decrease in eGFR from baseline, an approach previously proposed [25]. Measurements of renal function (serum creatinine) were obtained from follow-up visits after approximately 2, 4, 8, 13 and 21 years in the Steno-2 trial [26] and in the other observational cohort at follow-up visits after approximately 2, 3, 4 and 5 years. Furthermore, we considered additional renal endpoints and applied linear regression analysis to calculate rate of eGFR decline (eGFR slope) based on measurements from baseline, in individuals with at least two measurements and a minimum of follow-up of 3 years (n = 273). Participants from the Steno-2 trial (n = 138) had [51]chromium-ethylenediamine tetra-acetic acid ($^{51}$Cr-EDTA) clearance glomerular filtration rate (mGFR) measured by standard methods [27] at baseline and at follow-up visits. For the subset of individuals with mGFR, a subanalysis was performed for the endpoint of ≥30% decline in mGFR. In addition, an endpoint of ≥40% decline in eGFR from baseline was computed for a sensitivity analysis.

## Statistics

Baseline data are given as mean ± standard derivation (SD) for normal distributed continuous variables and as median and interquartile range (IQR) for non-normal distributed. Categorical variables are reported as numbers (%). The non-normal distributed variables, including the four metabolites, were log$_2$-transformed before correlation and regression analyses. We applied analyses of variance for normal distributed continuous variables and the Kruskal-Wallis analysis of variance for the non-normal distributed continuous variables; and the chi-square test for categorical variables to compare differences between quartiles of TMAO.

Correlations between the log$_2$-transformed four metabolites at baseline were examined using Pearson correlation and reported as the coefficient of determination ($R^2$).

Survival analyses with proportional hazard were performed to calculate hazard ratios (HRs) with 95% confidence intervals (CIs) per 1 SD increase in log$_2$-transformed concentrations of the metabolites for all endpoints. We considered all-cause mortality as a competing event for CVD and eGFR decline ≥30%, and non-CV mortality as a competing event for CV mortality. No other competing events were considered. We applied cause-specific hazards models when competing events were present. First, we performed unadjusted analyses. Next, we adjusted for baseline age, sex, HbA$_{1c}$, systolic blood pressure, body mass index, total cholesterol, smoking, UAER and eGFR by adding each variable solely and, thereafter, stepwise to the model. In addition to the analysis of the four metabolites separately, we calculated a weighted metabolite sum score as the weighted sum of concentrations of the four metabolites–a previously used method applied to reduce variability [28]. The weight for each metabolite was estimated as the parameter estimates for a 1 SD increase in the plasma concentrations of the metabolites, calculated from the proportional hazards model including all four metabolites and covariates. The weighted sum score was thereafter included as an exposure variable in the survival analyses. Moreover, we tested for heterogeneity in the HRs for the influence of age and sex by

introduction of the appropriate interaction terms in the proportional hazards model. We tested the proportional hazards model assumption for linearity of the logarithm of the metabolites for the endpoints by plotting parameter estimates of quintiles versus means of each quintile. Moreover, the assumption for proportional hazard was tested and fulfilled for all endpoints.

A two-tailed p-value<0.05 was considered significant. Statistical analyses were performed using SAS 9.4 software (SAS Institute, Cary, NC) and R (version 3.5.2; http://www.r-project.org/).

## Results

### Characteristics at baseline

Plasma TMAO and related metabolite concentrations were available from 311 individuals. Baseline clinical characteristics of the participants are summarized by quartiles of TMAO in **Table 1**. Overall, the mean±SD age was 57.2±8.2 years, known diabetes duration was median

**Table 1. Clinical characteristics at baseline.**

| | Overall | Trimethylamine N-oxide | | | | p |
| --- | --- | --- | --- | --- | --- | --- |
| | | Quartile 1 | Quartile 2 | Quartile 3 | Quartile 4 | |
| | | $\leq 3.8$ μM | > 3.8 to $\leq 5.9$ μM | > 5.9 to $\leq 9.0$ μM | > 9.0 μM | |
| Number of participants | 311 | 78 | 78 | 79 | 76 | |
| From the Steno-2 trial, n (%) | 138 | 48 (61.5) | 42 (53.8) | 28 (35.4) | 20 (26.3) | 0.003 |
| Age (years) | 57.2 ± 8.2 | 54.3 ± 8.1 | 56.6 ± 8.0 | 58.2 ± 8.2 | 59.8 ± 7.7 | <0.001 |
| Female, n (%) | 77 (24.8) | 20 (25.6) | 23 (29.5) | 15 (19.0) | 19 (25.0) | 0.50 |
| Known diabetes duration (years) | 8 (4–14) | 5 (3–10) | 8 (4–13) | 9 (5–14) | 10 (6–17) | 0.001 |
| Systolic blood pressure (mmHg) | 138 ± 20 | 141 ± 18 | 140 ± 22 | 133 ± 21 | 138 ± 20 | 0.038 |
| Diastolic blood pressure (mmHg) | 79 ± 12 | 83 ± 12 | 80 ± 13 | 76 ± 13 | 77 ± 10 | 0.002 |
| $HbA_{1c}$ (mmol/mol) | 64 ± 18.2 | 64 ± 18.8 | 65 ± 18.6 | 66 ± 19.0 | 62 ± 16.2 | 0.44 |
| $HbA_{1c}$ (%) | 8.0 ± 3.8 | 8.0 ± 3.9 | 8.1 ± 3.9 | 8.2 ± 3.9 | 7.8 ± 3.6 | 0.44 |
| Body mass index (kg/m$^2$) | 31.4 ± 5.5 | 30.6 ± 4.9 | 31.8 ± 5.7 | 31.9 ± 5.9 | 31.4 ± 5.5 | 0.45 |
| eGFR (ml/min/1.73m$^2$) | 93 ± 16.3 | 100 ± 14.8 | 96 ± 14.3 | 91 ± 17.0 | 86 ± 17.4 | <0.001 |
| UAER (mg/d) | 86 (51–180) | 90 (55–190) | 88 (54–177) | 90 (51–156) | 74 (37–173) | 0.33 |
| Serum LDL cholesterol (mM) | 2.5 ± 1.2 | 2.8 ± 1.2 | 2.8 ± 1.2 | 2.4 ± 1.2 | 2.2 ± 1.1 | 0.006 |
| Serum total cholesterol (mM) | 4.7 ± 1.4 | 4.9 ± 1.3 | 4.9 ± 1.5 | 4.6 ± 1.4 | 4.4 ± 1.4 | 0.10 |
| Current smoker, n (%) | 103 (33) | 32 (41) | 25 (32) | 23 (29) | 23 (30) | 0.38 |
| Plasma choline (μM) | 15.2 (12.4–19.2) | 12.7 (10.6–16.8) | 14.7 (12.5–18.2) | 16.3 (13.1–18.6) | 17.8 (14.0–21.7) | <0.001 |
| Plasma carnitine (μM) | 39.1 (31.0–49.9) | 35.3 (28.2–42.6) | 38.6 (30–48.2) | 39.8 (30.2–49.0) | 45.4 (35.9–57.0) | <0.001 |
| Plasma betaine (μM) | 35.0 (22.3–53.4) | 26.9 (18.7–43.2) | 31.8 (22.3–47.6) | 36.2 (25.0–58.9) | 41.5 (28.7–62.6) | <0.001 |
| Treatment with: | | | | | | |
| Oral antidiabetics, n (%) | 256 (83) | 60 (77) | 61 (78) | 68 (86) | 67 (88) | 0.17 |
| Insulin, n (%) | 118 (38) | 18 (23) | 28 (36) | 33 (42) | 39 (51) | 0.003 |
| RAAS blocking agents, n (%) | 197 (63) | 39 (50) | 44 (56) | 57 (72) | 57 (75) | 0.002 |
| Acetylsalicylic acid, n (%) | 178 (57) | 32 (41) | 45 (58) | 46 (58) | 55 (72) | 0.001 |
| Lipid lowering agents, n (%) | 165 (53) | 30 (39) | 34 (44) | 47 (60) | 54 (71) | <0.001 |

Data are shown as numbers (%), mean ± SD and median (IQR). eGFR: estimated glomerular filtrations rate; UAER: urinary albumin excretion rate; RAAS: renin-angiotensin-aldosterone system. We applied analyses of variance for normal distributed continuous variables and the Kruskal-Wallis analysis of variance for the non-normal distributed continuous variables; and the chi-square test for categorical variables to compare differences between quartiles of TMAO. The weighted metabolite sum score was calculated as the weighted sum of concentrations of the four metabolites.

(IQR) 8.0 (4.0–14.0) years and 24.8% were females. The plasma concentrations of TMAO were 5.9 (3.8–9.0) μM, choline 15.2 (12.4–19.2) μM, carnitine 39.1 (31.0–49.9) μM and betaine 35.0 (22.3–53.4) μM, respectively. Individuals with higher TMAO levels were older, had longer diabetes duration, lower eGFR, and lower LDL and total cholesterol levels than subjects with lower TMAO concentrations. In addition, at baseline individuals with higher TMAO were more frequently treated with insulin, renin-angiotensin-aldosterone-system blocking agents, cacetylsalicylic acid and cholesterol lowering agents.

Baseline concentrations of all four plasma compounds related to the TMAO pathway were correlated, and the strongest positive correlation was demonstrated between choline and carnitine; and between choline and betaine ($R^2$ = 0.33, p<0.0001 for both) (S1 Fig). TMAO was weakly positively correlated with the other metabolites (TMAO and choline: $R^2$ = 0.10; TMAO and carnitine: $R^2$ = 0.07; TMAO and betaine: $R^2$ = 0.08, p<0.001 for all).

## Mortality and CVD events

In the total population, follow-up was up to 21.9 years. Median (IQR) follow-up in years was 6.8 (6.1–15.5) for mortality and 6.5 (5.5–8.1) for CVD events. During follow-up a total of 106 persons died. Of 116 persons that suffered CVD events, 44 of these events were fatal.

The unadjusted and adjusted HRs with 95% CI for all endpoints are presented in **Tables 2 and 3** and visualised in **Fig 1**. In our study cohort, plasma TMAO was not associated with any of the endpoints. Higher plasma concentrations of choline and the weighted sum score of metabolites were in unadjusted analyses associated with all-cause mortality (HR [95% CI] 1.26 [1.01–1.57] and 1.09 [1.02–1.16], respectively). However, these associations were lost after adjustment. A higher weighted sum score was associated with cardiovascular mortality both before and after adjustment (adjusted HR 1.14 [1.00–1.29]). None of the metabolites were associated with risk of CVD.

## Progression of kidney disease

For the renal endpoint of an eGFR decline ≥30%, 289 subjects were available for analyses and 106 reached the endpoint over a median follow-up of 4.9 (3.7–8.0) years. Higher plasma

**Table 2. Plasma concentrations of TMAO and related compounds in relation to all-cause mortality, cardiovascular mortality and the combined cardiovascular endpoint.**

|  | All-cause mortality | p | CV mortality | p | CVD | p |
|---|---|---|---|---|---|---|
| Events, n (%) | 106 (34.1) |  | 44 (14.1) |  | 116 (37.3) |  |
| **TMAO**–unadjusted | 1.06 (0.88–1.28) | 0.53 | 1.09 (0.82–1.45) | 0.54 | 1.14 (0.96–1.35) | 0.14 |
| **TMAO**–adjusted | 1.02 (0.83–1.26) | 0.85 | 0.98 (0.70–1.37) | 0.92 | 1.11 (0.93–1.33) | 0.26 |
| **Choline**–unadjusted | **1.26 (1.01–1.57)** | **0.04** | 1.33 (0.95–1.87) | 0.10 | 0.98 (0.81–1.18) | 0.81 |
| **Choline**–adjusted | 1.17 (0.93–1.48) | 0.18 | 1.29 (0.89–1.86) | 0.18 | 0.98 (0.79–1.21) | 0.82 |
| **Carnitine**–unadjusted | 1.04 (0.85–1.28) | 0.70 | 1.28 (0.92–1.78) | 0.14 | 1.00 (0.83–1.21) | 0.96 |
| **Carnitine**–adjusted | 1.02 (0.82–1.27) | 0.86 | 1.26 (0.88–1.81) | 0.21 | 1.04 (0.84–1.27) | 0.75 |
| **Betaine**—unadjusted | 0.96 (0.77–1.19) | 0.70 | 0.92 (0.66–1.28) | 0.63 | 0.92 (0.75–1.12) | 0.38 |
| **Betaine**—adjusted | 0.97 (0.75–1.24) | 0.79 | 0.95 (0.63–1.41) | 0.78 | 1.02 (0.81–1.29) | 0.88 |
| **Weighted sum score**—unadjusted | **1.09 (1.02–1.16)** | **0.01** | **1.15 (1.03–1.29)** | **0.01** | 1.08 (0.98–1.20) | 0.12 |
| **Weighted sum score**—adjusted | 1.06 (0.99–1.14) | 0.09 | **1.14 (1.00–1.29)** | **0.04** | 1.08 (0.97–1.20) | 0.17 |

Proportional hazards models presented as hazard ratios (HR) with 95% confidence intervals for all-cause mortality, CV mortality and CVD. Cause-specific hazards models were applied for CV mortality and CVD. HRs express risk per 1 standard deviation increase in the log2 transformed plasma concentrations of the metabolites. Weighted sum score was calculated as a weighted score of the four metabolites. Adjustment included age, sex, HbA$_{1c}$, systolic blood pressure, body mass index, total cholesterol, smoking, urinary albumin excretion rate and eGFR at baseline. CVD: cardiovascular disease; eGFR: estimated glomerular filtration rate.

**Table 3. Plasma concentrations of TMAO and related compounds in relation to a decline in eGFR ≥30%, a decline in eGFR ≥40% and a decline in mGFR ≥30%.**

|  | eGFR decline ≥ 30% | p | eGFR decline ≥ 40% | p | mGFR decline ≥ 30% | p |
|---|---|---|---|---|---|---|
| Numbers available for analyses | 289 | | 289 | | 138 | |
| Events, n (%) | 106 (36.7) | | 78 (25.6) | | 89 (64.5) | |
| **TMAO–unadjusted** | 1.14 (0.97–1.37) | 0.12 | 1.04 (0.85–1.28) | 0.69 | 1.07 (0.89–1.27) | 0.47 |
| **TMAO–adjusted** | 1.08 (0.90–1.31) | 0.42 | 1.00 (0.80–1.27) | 0.96 | 1.11 (0.92–1.35) | 0.28 |
| **Choline–unadjusted** | **1.58 (1.27–1.96)** | **<0.001** | **1.69 (1.30–2.20)** | **<0.001** | **1.38 (1.04–1.83)** | **0.03** |
| **Choline–adjusted** | **1.49 (1.18–1.89)** | **0.001** | **1.66 (1.25–2.22)** | **<0.001** | **1.49 (1.08–2.04)** | **0.01** |
| **Carnitine–unadjusted** | 1.20 (0.98–1.47) | 0.08 | **1.42 (1.10–1.82)** | **0.006** | 1.08 (0.83–1.39) | 0.58 |
| **Carnitine–adjusted** | **1.27 (1.02–1.59)** | **0.03** | **1.55 (1.17–2.07)** | **0.003** | 1.31 (0.99–1.72) | 0.06 |
| **Betaine—unadjusted** | 1.10 (0.88–1.39) | 0.41 | 1.16 (0.89–1.52) | 0.28 | 0.96 (0.73–1.27) | 0.79 |
| **Betaine—adjusted** | 1.07 (0.82–1.39) | 0.63 | 1.14 (0.82–1.58) | 0.43 | 1.07 (0.78–1.47) | 0.69 |
| **Weighted sum score—unadjusted** | **1.40 (1.20–1.64)** | **<0.001** | **1.49 (1.23–1.81)** | **<0.001** | **1.28 (1.02–1.61)** | **0.04** |
| **Weighted sum score—adjusted** | **1.36 (1.16–1.60)** | **<0.001** | **1.54 (1.25–1.89)** | **<0.001** | **1.49 (1.13–1.97)** | **0.005** |

Cause-specific proportional hazards models presented as hazard ratios (HR) with 95% confidence intervals for decline in eGFR ≥30%, decline in eGFR ≥40% and decline in mGFR ≥30%. HRs express risk per 1 standard deviation increase in the log2 transformed concentrations of the metabolites. Weighted sum score was calculated as a weighted score of the four metabolites. Adjustment included baseline age, sex, HbA$_{1c}$, systolic blood pressure, body mass index, total cholesterol, smoking, urinary albumin excretion rate and eGFR/mGFR, as appropriate. eGFR: estimated glomerular filtration rate; mGFR: measured glomerular filtration rate.

choline and the weighted sum score were risk markers for a decline in eGFR ≥30%, in both unadjusted and adjusted models. Higher plasma concentrations of carnitine were also a risk marker in the adjusted model (HR 1.27 [1.02–1.59]), but not unadjusted (p = 0.08) (Table 3).

We also evaluated the risk of an eGFR decline ≥40% (78 events) presented in Table 3. Higher plasma concentrations of choline, carnitine and the weighted sum score were associated with higher risk of eGFR decline ≥40% both unadjusted and after adjustment (adjusted HR choline: 1.66 [1.25–2.22], carnitine: 1.55 [1.17–2.07] and weighted sum score: 1.54 [1.25–1.89]).

In a regression model with yearly change in eGFR (n = 273) evaluated over a median of 5.0 (4.0–13.0) years, higher concentrations of choline were associated with a steeper yearly decline in eGFR in both unadjusted (p = 0.001) and adjusted (p<0.001) models. Higher carnitine levels were also associated with a steeper yearly decline in eGFR in adjusted analyses (p = 0.04). Plasma TMAO and betaine were not associated with yearly change in eGFR (p≥0.18).

In the subset of participants from the Steno-2 trial with measurements of [51]Cr-EDTA GFR (mGFR) at baseline and during follow-up (n = 138), we estimated the risk of mGFR decline ≥30% (89 events). In these analyses, higher plasma concentrations of choline and the weighted sum score were risk markers both in unadjusted and adjusted models (adjusted HR 1.49 [1.08–2.06] and 1.49 [1.13–1.96], respectively), Table 3. TMAO and betaine were not associated with mGFR decline ≥30%.

Additional inclusion of the plasma concentrations of TMAO in the proportional hazards models for choline, carnitine and betaine did not change the overall results for any of the endpoints (mortality, CVD events or renal endpoints).

## Additional analyses

In sensitivity analyses, we included additional adjustment for cohort and treatment with RAAS blocking agents, insulin, acetylsalicylic acid and lipid lowering agents. This did not change any of the results. To consider competing risk for the endpoint of CV mortality, a proportional hazard model was performed for non-CV mortality (n = 62); None of the

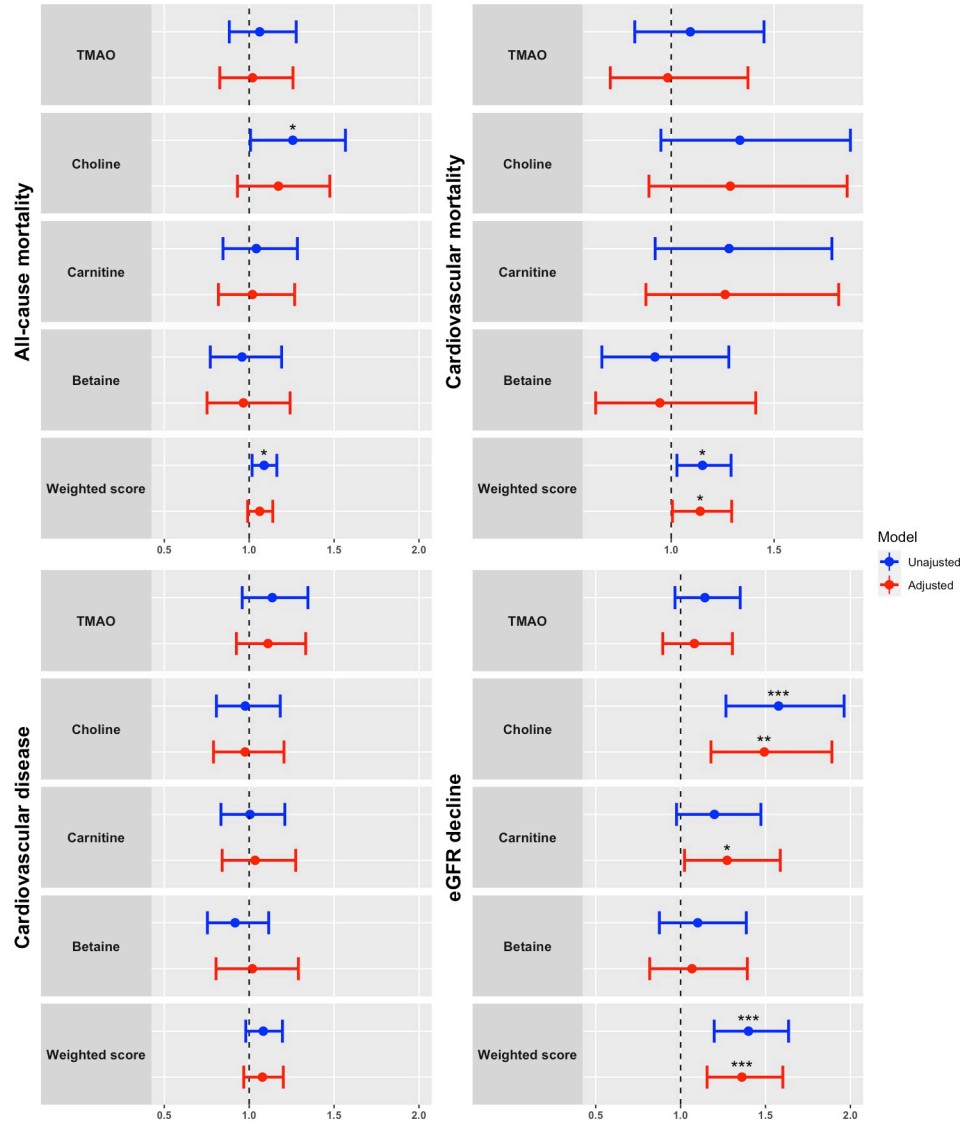

**Fig 1. Forrest plot visualizing of plasma concentrations of TMAO and related compounds in relation to clinical endpoints.** Proportional hazards models presented as hazard ratios (HRs) with 95% confidence intervals for all-cause mortality (n = 106), CV mortality (n = 44), CVD (n = 116) and eGFR decline ≥30% (n = 106). HRs express risk per 1 standard deviation increase in the log2 transformed plasma concentrations of the metabolites. Weighted sum score was calculated as a weighted score of the four metabolites. Adjustment included age, sex, $HbA_{1c}$, systolic blood pressure, body mass index, serum total cholesterol, smoking, urinary albumin excretion rate and eGFR at baseline. Level of significance is denoted as follows: *: $0.01 \leq p < 0.05$, **: $0.001 \leq p < 0.01$; ***: $p < 0.001$. TMAO = trimethylamine N-oxide.

metabolites or weighted metabolite sum score were associated with the risk of non-CV mortality ($p \geq 0.14$).

## Discussion

We investigated whether plasma concentrations of TMAO and its metabolic precursors choline, carnitine and betaine as well as a composite score of metabolites, were associated with mortality, adverse CVD events and deterioration in renal function in individuals with T2D and albuminuria.

We found that higher plasma concentrations of choline and carnitine (carnitine, only in adjusted models) were directly associated with deterioration in renal function. Also, when we combined the four examined metabolites in the TMAO pathway into a weighted sum score, a higher score was positively associated with deterioration in renal function. Furthermore, the weighted sum score of metabolites was directly associated with cardiovascular mortality. We could not demonstrate any association between plasma concentrations of TMAO or its metabolic precursors and the risk of CVD.

The TMAO pathway is the first in the literature that links metabolites produced by the gut microbiota to the risk of cardiovascular and renal diseases especially in cohorts with established CVD [3, 4, 7]. Such associations may be of interest in the search for new therapeutic targets for prevention of CVD and progression of renal disease.

In our study we found an association between the weighted sum score and CV mortality, but not when the metabolites were analysed separately. This may indicate that calculating a weighted sum of the TMAO related metabolites is justified potentially by reducing variability in the measurements and day-to-day variability of individual metabolites. In a previous report a comparable sum score including five metabolites in the TMAO pathway (TMAO, choline, phosphocholine, α-glycerophosphocholine, and betaine) was calculated and shown to be linked to an increased risk of CVD events [28].

Previous studies in persons with T2D have demonstrated a positive association between higher plasma concentrations of TMAO and increased risk of CVD events and mortality even after adjustment for traditional CVD risk factors [6, 8, 29, 30]. In addition, in recent meta-analyses, higher concentrations of TMAO were reported to be a risk marker of mortality and CVD risk amongst all subjects [7]. In our cohort, we could not replicate such associations which may be explained by clinical differences between the reported cohorts and ours. Previous studies in T2D included individuals undergoing elective coronary angiography or with already existing CVD at baseline. In the present study, only a few of our included participants had known coronary artery disease at baseline as it was an exclusion criterion for the individuals enrolled in our study during 2007–08. Moreover, the reported studies had a lower median eGFR ranging between 62–82 (50–94) ml/min/1.73 m$^2$ [6, 30] compared to our participants with median eGFR 96 (82–104) ml/min/1.73 m$^2$. Still, the impact of renal function as a potential mediator or contributing factor of the adverse effect of TMAO is debated [12, 15, 31]. Though it should be noted that in a recent meta-analysis that examined the impact of renal function on TMAO associations with all-cause mortality and CVD, TMAO remained significantly associated with both all-cause mortality and CVD risks in individuals with and without CKD [7]. The level of measured plasma TMAO in our study did not differ substantially from previous studies (5.9 [3.8–9.0] μM in our study compared with 4.4 [2.8–7.7] μM and 7.5 [4.4–12.1] μM, respectively in the other studies [6, 30]). Another limitation to consider is the lack of statistical power. However, due to the long duration of our study (up to 21 years), we have many person-years of follow-up and a relatively high overall event rate in the aging population monitored. Further, we have with previous markers that show strong association with renal function like plasma symmetric and asymmetric dimethylarginine and urinary kidney injury molecule 1 (u-KIM-1), shown an association between these and cardiovascular disease [19, 32, 33].

Consistent with our study, a recent paper also failed to show an association between TMAO and CVD in T2D [34]. This study by Cardona *et al.* was a post-hoc case-control analyses of the ACCORD study including persons with T2D without evident atherosclerotic disease at baseline. With similar plasma concentrations of TMAO and traditional CVD risk factors at baseline between controls and cases the investigators concluded that the lack of a link between plasma concentration of TMAO and CVD might be due to the absence in difference of

traditional CVD risk factors. Another study including individuals with high cardiovascular risk when estimated from conventional CVD risk factors, but free of CVD at baseline also failed to show an association with incident CVD, indicating that plasma TMAO concentration may be a better risk marker in established CVD [28]. In line with this, a very recent larger study of a T2D cohort found an association between higher plasma concentrations of TMAO and cardiovascular endpoints and mortality, however these associations were insignificant in subanalyses excluding participants with known CVD at baseline [8]. It is also of interest to note that a recent experimental and human study indicated that TMAO might be a better marker of instability of established atherosclerotic plaques, than the development of or extent of atherosclerosis [35].

It is worth noticing that in our cohort the individuals with the highest plasma concentrations of TMAO also were those with the highest risk profile of traditional CVD risk factors. Drugs intended to lower these risk factors were also more frequently prescribed in the individuals with highest plasma TMAO levels. This may have attenuated a potential association between TMAO and the outcomes. In this situation it would be difficult to conclude, whether plasma concentrations of TMAO (and its precursors) are a risk factor for CVD or just markers of a high-risk profile based on known risk factors.

The results of previous studies on the association between the other metabolites of the TMAO pathway and risk of mortality and CVD have been inconsistent [4, 9, 14, 28, 36]. In a large population-based study, higher plasma concentration of phosphatidylcholine was directly associated with risk of all-cause mortality, especially CVD related [36]. An association that was stronger among individuals with diabetes than in those without. In contrast, the association between plasma concentrations of choline and betaine and the risk of CVD may be confounded by the plasma level of TMAO as the main driver of the association [14]. In our study, the associations for choline were independent of the level of TMAO and a recent meta-analysis could not demonstrate any association between circulating concentrations of choline or betaine and risk of CVD [37].

A few studies have investigated the association between circulating choline and renal outcome in humans indicating a positive association between choline and an aggravation of renal function decline [9, 38]. In addition, experimental studies in a mouse model have demonstrated that TMAO and choline are associated with renal damage expressed as increased circulating concentrations of Cystatin C and KIM-1 [3].

Potential therapeutic targets of the TMAO pathway have recently been investigated in experimental murine models with promising results. In a mice model of chronic kidney disease, small molecule inhibition of microbial TMA generation (the precursor of TMAO) with a mechanism-based inhibitor iodomethylcholine (a TMA-lyase inhibitor predominately targeting TMA production in the gut lumen) reduced the decline in renal function and tubulointerstitial fibrosis [39]. A general inhibition of the same pathway also attenuates atherosclerosis and thrombosis in mice [40, 41]. The molecular mechanisms of TMAO's adverse effect on kidney impairment has been shown to involve TGF-β induced signalling pathways, and induction of multiple pro-fibrosis forming gene expression changes, and yet the receptor involved in this effect remains unknown and needs further exploration [39]. The composition of diet may also be a regulating factor of TMAO and the impact of CVD and renal impairment. A large observational study (n = 55,113) demonstrated that vegans (e.g. a low choline diet) had a lower prevalence of chronic kidney disease compared to omnivores [42]. Diet restriction may therefore also be considered as an intervention. However, randomised controlled clinical trials are needed to clarify any causal effects of these interventions.

Our study has limitations. The plasma metabolites were not measured in the fasting state and only measured once at baseline and hence day-to-day variability due to for example

expected variation in diet might have influenced our results. Another limitation is that the metabolites were measured in samples stored for a long period. Even though plasma concentrations of TMAO show stability when stored on -80°C over 5 years with multiple freeze thraw cycles [43], stability of the metabolites are not known for follow-up of up to 21 years. Medication use throughout follow-up period was not incorporated into any analyses presented. Moreover, it is possible that estimation of GFR using cystatin C would have increased accuracy, however we do not expect this would have affected our findings significantly [44]. Finally, the study population included 77% males and although we did not find any significant effect modification on sex, there could be variations in the TMAO pathway with sex, which have impacted our results. Strengths of our study include a well-characterized cohort and complete long-time follow-up based on data from Danish registries with high quality.

In conclusion, higher plasma concentrations of choline, carnitine and a weighted sum of the concentrations of four plasma metabolites related to the TMAO pathway were directly associated with a deterioration of renal function in Danish white individuals with T2D and albuminuria. Moreover, a higher weighted sum score of the TMAO pathway-related metabolites was positively associated with risk of CV mortality, whereas individual metabolites were not.

## Supporting information

**S1 Fig. Pearson correlations between the four log$_2$-transformed plasma metabolites.** Numbers are $R^2$ and all p-values <0.0001. TMAO = trimethylamine N-oxide.
(TIFF)

## Acknowledgments

We thank the lab technicians at Steno Diabetes Center Copenhagen for technical assistance.

## Author Contributions

**Conceptualization:** Hans-Henrik Parving, Oluf Pedersen, Peter Rossing.

**Data curation:** Jens Christian Øllgaard, Bernt Johan von Scholten, Henrik Reinhard, Zeneng Wang, Peter Gæde, Oluf Pedersen.

**Formal analysis:** Signe Abitz Winther, Jens Christian Øllgaard, Tine Willum Hansen, Tarunveer Singh Ahluwalia, Stanley Hazen.

**Investigation:** Oluf Pedersen, Peter Rossing.

**Methodology:** Signe Abitz Winther, Oluf Pedersen.

**Supervision:** Tine Willum Hansen, Bernt Johan von Scholten, Tarunveer Singh Ahluwalia, Stanley Hazen, Oluf Pedersen, Peter Rossing.

**Validation:** Signe Abitz Winther.

**Visualization:** Signe Abitz Winther, Tarunveer Singh Ahluwalia.

**Writing – original draft:** Signe Abitz Winther.

**Writing – review & editing:** Signe Abitz Winther, Jens Christian Øllgaard, Tine Willum Hansen, Bernt Johan von Scholten, Henrik Reinhard, Tarunveer Singh Ahluwalia, Zeneng Wang, Peter Gæde, Hans-Henrik Parving, Stanley Hazen, Oluf Pedersen, Peter Rossing.

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
