## [Decision Letter · Decision Letter 0]

26 Aug 2020

PONE-D-20-21640

Plasma trimethylamine N-oxide and its metabolic precursors and risk of mortality, cardiovascular and renal disease in individuals with type 2-diabetes and albuminuria

PLOS ONE

Dear Dr. Winther,

Thank you for submitting your manuscript to PLOS ONE. After careful consideration, we feel that it has merit but does not fully meet PLOS ONE’s publication criteria as it currently stands. Therefore, we invite you to submit a revised version of the manuscript that addresses the points raised during the review process.

Please carefully address the points raised by the reviewers and especially focus on: inclusion criteria, statistical analyses employed and the cystatin C issue raised.  

We look forward to receiving your revised manuscript.

Kind regards,

M. Faadiel Essop

Academic Editor

PLOS ONE

Journal Requirements:

2. Please ensure you have included the registration number for the clinical trial referenced in the manuscript.

"I have read the journal's policy and the authors of this manuscript have the following competing interests:

PR reports having given lectures for Astra Zeneca, Bayer, Novo Nordisk and Boehringer Ingelheim, and has served as a consultant for AbbVie, Astra Zeneca, Bayer, Eli Lilly, Boehringer Ingelheim, Astellas, Gilead, Mundipharma, Vifor, and Novo Nordisk, all fees given to Steno Diabetes Center Copenhagen. SAW, TWH, BJvS, TSA and PR own stocks in Novo Nordisk A/S and TSA in Zealand Pharma A/S. ZW and SLH are named as co-inventors on pending and issued patents held by the Cleveland Clinic relating to cardiovascular diagnostics and therapeutics, and have the right to receive royalty payment for inventions or discoveries related to cardiovascular diagnostics or therapeutics from Cleveland Heart Lab, Quest Diagnostics and Proctor & Gamble. SLH also reports having been paid as a consultant from Proctor & Gamble, and having received research funds from Proctor & Gamble and Roche. The other authors report no conflicts of interest.  "

We note that one or more of the authors are employed by a commercial company: Novo Nordisk A/S, Denmark.

3.1. Please provide an amended Funding Statement declaring this commercial affiliation, as well as a statement regarding the Role of Funders in your study. If the funding organization did not play a role in the study design, data collection and analysis, decision to publish, or preparation of the manuscript and only provided financial support in the form of authors' salaries and/or research materials, please review your statements relating to the author contributions, and ensure you have specifically and accurately indicated the role(s) that these authors had in your study. You can update author roles in the Author Contributions section of the online submission form.

3.2. Please also provide an updated Competing Interests Statement declaring this commercial affiliation along with any other relevant declarations relating to employment, consultancy, patents, products in development, or marketed products, etc.  

6. Please include your tables as part of your main manuscript and remove the individual files. Please note that supplementary tables (should remain/ be uploaded) as separate "supporting information" files.

Reviewers' comments:

Reviewer's Responses to Questions

**Comments to the Author**

1. Is the manuscript technically sound, and do the data support the conclusions?

Reviewer #1: Yes

Reviewer #2: Yes

Reviewer #3: Partly

Reviewer #4: Yes

2. Has the statistical analysis been performed appropriately and rigorously? 

Reviewer #1: Yes

Reviewer #2: Yes

Reviewer #3: Yes

Reviewer #4: Yes

3. Have the authors made all data underlying the findings in their manuscript fully available?

Reviewer #1: Yes

Reviewer #2: No

Reviewer #3: No

Reviewer #4: Yes

4. Is the manuscript presented in an intelligible fashion and written in standard English?

Reviewer #1: Yes

Reviewer #2: Yes

Reviewer #3: Yes

Reviewer #4: Yes

5. Review Comments to the Author

Reviewer #1: In this study investigators look at possible associations between plasma levels of trimethylamine-N-oxide and its metabolic precursors, and CVD and renal disease in patients with type 2 diabetic and albuminuria. This metabolic pathway and the role its metabolites may play in disease is still relatively underexplored. The outcomes of the studies that have been published appear to be relatively inconclusive and controversial. This study uses long term data from a reasonably large patient cohort to come to the conclusion that choline, carnitine and the weighted sum of the TMAO pathway metabolites may be risk markers for deteriorating renal function. They also show that only a higher weighted sum score of TMAO pathway metabolites shows an association with increased risk of CV mortality while individual metabolites alone do not. The manuscript is well written and data analyses appear to be well executed. The authors’ conclusions are reasonable and supported by the data.

Minor comments

1. The sentence on line 198 should be rephrased. Perhaps “In addition, at baseline individuals with higher TMAO………”

2. Line 208 onwards. Last sentence in particular is difficult to follow. “During follow-up a total of 106 patients died. Of 116 patients that suffered CVD events, 44 of these events were fatal.” Perhaps?

3. Line 289. “ Consistent with our study……".

Reviewer #2: I have to commend the authors on a well documented and high impact research. The problem statement was complimented with a focussed introduction and provided the reader with sufficient background leading to the motivation and rationale for the current study. A large cohort of patients that participated in 2 clinical trials were recruited for this study. Four plasma metabolites that are linked to the TMAO pathway was investigated with respect to their risk with all-cause mortality, CVD and deterioration in n=311 renal function in individuals with type 2-diabetes (T2D) and albuminuria.

Appropriate and sensitive measurements were performed to quantify the variables.

I would like to know what the inclusion criteria was for this study? This is not clearly described.

All other aspects of the manuscript is in my opinion flawless. I was curious as to why cystatin C was not estimated in this study? Not much is reported on this but I was just curious.

I really enjoyed the variety of figures and tables that clearly indicated the contribution of the biological markers toward risk of mortality,cardiovascular and renal disease.

Reviewer #3: Here is a list of specific comments. Note: line and page numbering in reviews and comments is based on ruler applied in Editorial Manager-generated PDF.

1. Page 7, line 141: In the Endpoints section, please make sure all four endpoints, all-cause mortality, CV mortality, CVD and eGFR decline >=30%, for survival analyses were properly introduced.

2. Page 7, lines 148–149: “Censored” might not be an appropriate word here. I suggest writing ‘individuals experiencing multiple events stopped their follow up at time of the first event in the analyses’.

3. Page 8, lines 165–166: Please clarify “analyses” referred to ‘correlation and regression analyses’, not including summary statistics in Table 1.

4. Page 8, line 166: For the non-normal continuous variables (those reported in median and IQR), please use a non-parametric version of analysis of variance (e.g., Kruskal-Wallis analysis of variance).

5. Page 8, lines 170–172: For all-cause mortality, Cox proportional hazards models were appropriate. For CV mortality, CVD and eGFR decline >=30%, please describe whether or not there were competing events for each endpoint. If competing events existed, the models in Table 2 were technically called cause-specific hazards models (by censoring the competing events). You do not have to change the analyses in Table 2, but, for each endpoint, clarify the presence/absence of competing events and the use of cause-specific hazards models when the competing events present.

6. Page 8, lines 173–174: Please clarify if these variables were the candidates of the forward selection approach or the results of the selection. Table 2 suggested the latter. For the latter, how many and what variables did you include at the beginning of the forward selection? Different endpoints should result in different sets of selections. How did you determine the final adjusted covariates for all endpoints? Please specify responses to the above questions in this paragraph. Please comment on why treatments and cohorts were not included; I would suggest including them even if they did not pass the inclusion criteria of the forward selection approach.

7. Page 8, line 179: Because of the competing events and the use of cause-specific hazards models (see Comment #5 above), I suggest replacing “Cox regression model” with ‘proportional hazards models’ such that both Cox and cause-specific hazards models are included. Apply this accordingly for the rest of the manuscript.

8. Page 9, line 203: Please confirm the correlation of 0.33 were r or r-squared. The Statistics section described the use of Pearson correlation. I assumed it was r, not r-squared.

9. Page 9, line 203: I suggest including p-values in S1 Fig.

10. Page 10, line 220: Please include this information in Table 2.

11. Page 11, line 226: I suggest combining eGFR decline >=30% with S1 Table into Table 3. If it exceeds the limit of tables/figures, I would move Fig. 1 to S2 Fig.

12. Table 1: I suggest including cohorts and the weighted metabolite sum score in the table.

13. Table 2: For each endpoint, if the sample size was not 311, please specify in the table.

Reviewer #4: Review on the manuscript entitled: “Plasma trimethylamine N-oxide and its metabolic precursors and risk of mortality, cardiovascular and renal disease in individuals with type 2-diabetes and albuminuria” by Winther SA et al

The manuscript’s aim is to investigate the associations between four plasma metabolites in the TMAO pathway and risk of all-cause mortality, cardiovascular disease, and deterioration in renal function in individuals with type 2 diabetes and albuminuria. Levels of plasma metabolites were measured at baseline in individuals with type 2 diabetes, while other information was obtained from registries. Associations were analysed using Cox models. The results showed that in type 2 diabetes and albuminuria, higher choline, carnitine, and a weighted sum of four metabolites from the TMAO pathway were risk markers for deterioration in renal function in the follow up.

This is a beautifully written manuscript with all aspects very well clarified. I find it easy to follow from the title, the abstract, and the rest of the article. The authors did address all their findings, and also highlighting the limitations of the study accordingly.

Something minor I noticed is that in their subjects 77% were females. Was wondering if perhaps there were any variations in the TMAO pathway with gender of the patients?

6. PLOS authors have the option to publish the peer review history of their article (what does this mean?). If published, this will include your full peer review and any attached files.

Reviewer #1: **Yes: **Eugene du Toit

Reviewer #2: No

Reviewer #3: No

Reviewer #4: No

---

## [Author Response · Author response to Decision Letter 0]

2 Nov 2020

Reviewer #1: 

In this study investigators look at possible associations between plasma levels of trimethylamine-N-oxide and its metabolic precursors, and CVD and renal disease in patients with type 2 diabetic and albuminuria. This metabolic pathway and the role its metabolites may play in disease is still relatively underexplored. The outcomes of the studies that have been published appear to be relatively inconclusive and controversial. This study uses long term data from a reasonably large patient cohort to come to the conclusion that choline, carnitine and the weighted sum of the TMAO pathway metabolites may be risk markers for deteriorating renal function. They also show that only a higher weighted sum score of TMAO pathway metabolites shows an association with increased risk of CV mortality while individual metabolites alone do not. The manuscript is well written and data analyses appear to be well executed. The authors’ conclusions are reasonable and supported by the data.

Minor comments

1. The sentence on line 198 should be rephrased. Perhaps “In addition, at baseline individuals with higher TMAO………”

Thank you for this helpful comment. The sentence has been rephrased as suggested (revised version: Results, line 207).

2. Line 208 onwards. Last sentence in particular is difficult to follow. “During follow-up a total of 106 patients died. Of 116 patients that suffered CVD events, 44 of these events were fatal.” Perhaps?

We agree and have rephrased the sentence as suggested (revised version: Results, line 220-221)

3. Line 289. “ Consistent with our study……".

Thank you, we have corrected as suggested (revised version: Discussion, line 316)

Reviewer #2: 

I have to commend the authors on a well documented and high impact research. The problem statement was complimented with a focussed introduction and provided the reader with sufficient background leading to the motivation and rationale for the current study. A large cohort of patients that participated in 2 clinical trials were recruited for this study. Four plasma metabolites that are linked to the TMAO pathway was investigated with respect to their risk with all-cause mortality, CVD and deterioration in n=311 renal function in individuals with type 2-diabetes (T2D) and albuminuria.

Appropriate and sensitive measurements were performed to quantify the variables.

I would like to know what the inclusion criteria was for this study? This is not clearly described.

Thank you for this helpful comment. We have clarified the inclusion criteria for the two cohorts in the revised Methods:

Page 5; lines 103-104: 

 “Inclusion criteria included individuals with T2D (defined according to the 1985 WHO criteria), age 40-65 years and persistent microalbuminuria.”

Page 5; lines 107-109: 

 “The second cohort comprised 200 T2D individuals with persistent albuminuria (two out of three consecutive measured UAER > 30 mg/24h). Other inclusion criteria included age between 20 and 70 years, no known coronary artery disease and normal plasma creatinine. Of the 200 participants included in the original observational study, 173 had available metabolite measures for the present study”.

For the present study the inclusion criteria were available metabolite measurements. This is now specified in the revised Methods:

Page 6; lines 115-116:

“Individuals from the original two studies with available metabolite measures from stored blood samples were included in the present study.”

All other aspects of the manuscript is in my opinion flawless. I was curious as to why cystatin C was not estimated in this study? Not much is reported on this but I was just curious.

We agree that estimation of cystatin C might have been a helpful supplement to the analyses. Unfortunately, cystatin C was not measured in the study. We only include cystatin C in studies with for example weight loss or with medications affecting creatinine clearance. However, we agree that cystatin C or the combination of creatinine and cystatin C may improve precision when estimating GFR.

The lack of information on cystatin C is included in the revised limitation section:

Page 23; lines 373-377): 

Moreover, it is possible that estimation of GFR using cystatin C would have increased accuracy, however we do not expect this would have affected our findings significantly (Michael G Shlipak et al. N Engl J Med 2013; 69:932-43).

I really enjoyed the variety of figures and tables that clearly indicated the contribution of the biological markers toward risk of mortality, cardiovascular and renal disease.

Reviewer #3: Here is a list of specific comments. Note: line and page numbering in reviews and comments is based on ruler applied in Editorial Manager-generated PDF.

1. Page 7, line 141: In the Endpoints section, please make sure all four endpoints, all-cause mortality, CV mortality, CVD and eGFR decline >=30%, for survival analyses were properly introduced.

Thank you for pointing this out. We have introduced all four endpoints in the revised Endpoints section: 

Page 7; lines 148-149): 

“We considered four endpoints; All-cause mortality, CV mortality, CVD and eGFR decline ≥30%”.

And have clarified the definition of CV mortality:

Page 7; lines 149-150: 

“All deaths were classified as CVD related unless an unequivocal noncardiovascular cause was reported, a previously recognized approach (Yusuf S et al. N Engl J Med 2000;342:154–160)”.

Moreover, we have clarified that we considered additional renal endpoints: 

Page 8; lines 158-160: 

“Furthermore, we considered additional renal endpoints and applied linear regression analysis to calculate rate of eGFR decline (eGFR slope) based on measurements from baseline, in individuals with at least two measurements and a minimum of follow-up of 3 years (n = 273)”.

2. Page 7, lines 148–149: “Censored” might not be an appropriate word here. I suggest writing ‘individuals experiencing multiple events stopped their follow up at time of the first event in the analyses’.

Thank you for this suggestion. We have reworded as proposed (revised version: Results, Endpoints section, page 7; lines 152-153).

3. Page 8, lines 165–166: Please clarify “analyses” referred to ‘correlation and regression analyses’, not including summary statistics in Table 1.

Thank you for the comment. We have clarified that the “analyses” referred to ‘correlation and regression analyses (revised version: Statistics, page 8; line 171).

4. Page 8, line 166: For the non-normal continuous variables (those reported in median and IQR), please use a non-parametric version of analysis of variance (e.g., Kruskal-Wallis analysis of variance).

In accordance with this comment we now apply the Kruskal-Wallis analysis of variance for the non-normal distributed continuous variables in the revised Statistics: 

Page 8; lines 171-173: 

“We applied analyses of variance for normal distributed continuous variables and the Kruskal-Wallis analysis of variance for the non-normal distributed continuous variables; and the chi-square test for categorical variables to compare differences between quartiles of TMAO”.

Table 1 is revised accordantly.

5. Page 8, lines 170–172: For all-cause mortality, Cox proportional hazards models were appropriate. For CV mortality, CVD and eGFR decline >=30%, please describe whether or not there were competing events for each endpoint. If competing events existed, the models in Table 2 were technically called cause-specific hazards models (by censoring the competing events). You do not have to change the analyses in Table 2, but, for each endpoint, clarify the presence/absence of competing events and the use of cause-specific hazards models when the competing events present.

Thank you for this very relevant comment. All-cause mortality is a competing event for CVD and eGFR decline ≥30%, and non-CV mortality is a competing event for CV mortality. No other competing events were considered in the analyses. This is clarified in the revised Statistics: 

Page 9; lines 179-181: 

“We considered all-cause mortality as a competing event for CVD and eGFR decline ≥30%, and non-CV mortality as a competing event for CV mortality. No other competing events were considered”. 

As non-CV mortality was a competing event for CV mortality, we have included analyses of non-CV mortality in a new added section in the revised version (Additional analyses): 

Page 18-19; lines 268-270: 

“To consider competing risk for the endpoint of CV mortality, a proportional hazard model was performed for non-CV mortality (n=62); None of the metabolites or weighted metabolite sum score were associated with the risk of non-CV mortality (p≥0.14).”

Moreover, we have clarified that we applied cause-specific hazards models when the competing events were present. Revised statistical section: 

Page 9; lines 181-182: 

“We applied cause-specific hazards models when competing events were present.”

Legends to Table 2 and new Table 3 (former Table S1) are also updated accordantly.

6. Page 8, lines 173–174: Please clarify if these variables were the candidates of the forward selection approach or the results of the selection. Table 2 suggested the latter. For the latter, how many and what variables did you include at the beginning of the forward selection? Different endpoints should result in different sets of selections. How did you determine the final adjusted covariates for all endpoints? Please specify responses to the above questions in this paragraph. Please comment on why treatments and cohorts were not included; I would suggest including them even if they did not pass the inclusion criteria of the forward selection approach.

Thank you for the comment, which gives us the opportunity to clarify. The mentioned variables were selected for all endpoint based on their clinical relevance for the studied cohort. After running the unadjusted regression models each variable were added stepwise to the model. This is now specified in the revised version:

Page 9; lines 182-184: 

“First, we performed unadjusted analyses. Next, we adjusted for baseline age, sex, HbA1c, systolic blood pressure, body mass index, total cholesterol, smoking, UAER and eGFR by adding each variable solely and, thereafter, stepwise to the model.”

Furthermore, as suggested, we have performed additional adjustment for cohort and treatment with RAAS blocking agents, insulin, acetylsalicylic acid and lipid lowering agents included in a new section in the revised manuscript named Additional analyses. 

Page 18; lines 266-268: 

“In sensitivity analyses, we included additional adjustment for cohort and treatment with RAAS blocking agents, insulin, acetylsalicylic acid and lipid lowering agents. This did not change any of the results”.

7. Page 8, line 179: Because of the competing events and the use of cause-specific hazards models (see Comment #5 above), I suggest replacing “Cox regression model” with ‘proportional hazards models’ such that both Cox and cause-specific hazards models are included. Apply this accordingly for the rest of the manuscript.

Thank you for this helpful suggestion. We have replaced “Cox regression model” with ‘proportional hazards models’ throughout the manuscript.

8. Page 9, line 203: Please confirm the correlation of 0.33 were r or r-squared. The Statistics section described the use of Pearson correlation. I assumed it was r, not r-squared.

Thank you for noticing this inconsistence. The correlations presented are r-squared. This is corrected in the statistical section: 

Page 8; lines 175-176: 

“Correlations between the log2-transformed four metabolites at baseline were examined using Pearson correlation and reported as the coefficient of determination (R2).”

9. Page 9, line 203: I suggest including p-values in S1 Fig.

As suggested, p-values are added to the legend of the S1 Fig.

10. Page 10, line 220: Please include this information in Table 2.

In accordance with your comment number 11, we have combined eGFR decline ≥30% with S1 Table into Table 3. Thus, the requested information on numbers of subjects available for analyses are included in the revised Table 3 for all the renal endpoints.

11. Page 11, line 226: I suggest combining eGFR decline >=30% with S1 Table into Table 3. If it exceeds the limit of tables/figures, I would move Fig. 1 to S2 Fig.

Thank you for this valuable suggestion. We have combined eGFR decline ≥30% with the other renal endpoints into a Table 3 instead of the former Table S1. The manuscript is updated accordantly.

12. Table 1: I suggest including cohorts and the weighted metabolite sum score in the table.

As suggested, we now include information on cohort in Table 1. The weighted metabolite sum score is calculated specific for each endpoint and also include negative values due to the calculated different influence of each metabolite. Thus, the absolute numbers of the weighted metabolite sum score would not be clinically relevant. Therefore, we decided not to include the weighted metabolite sum score in Table 1. 

13. Table 2: For each endpoint, if the sample size was not 311, please specify in the table.

For all the endpoints presented in the updated Table 2 the sample size was 311. However, for the renal endpoints presented in the revised Table 3 the sample size was not 311. Thus, we have specified the sample size for each of the outcomes presented in Table 3.

Reviewer #4: Review on the manuscript entitled: “Plasma trimethylamine N-oxide and its metabolic precursors and risk of mortality, cardiovascular and renal disease in individuals with type 2-diabetes and albuminuria” by Winther SA et al

The manuscript’s aim is to investigate the associations between four plasma metabolites in the TMAO pathway and risk of all-cause mortality, cardiovascular disease, and deterioration in renal function in individuals with type 2 diabetes and albuminuria. Levels of plasma metabolites were measured at baseline in individuals with type 2 diabetes, while other information was obtained from registries. Associations were analysed using Cox models. The results showed that in type 2 diabetes and albuminuria, higher choline, carnitine, and a weighted sum of four metabolites from the TMAO pathway were risk markers for deterioration in renal function in the follow up.

This is a beautifully written manuscript with all aspects very well clarified. I find it easy to follow from the title, the abstract, and the rest of the article. The authors did address all their findings, and also highlighting the limitations of the study accordingly.

Something minor I noticed is that in their subjects 77% were females. Was wondering if perhaps there were any variations in the TMAO pathway with gender of the patients?

Thank you for the comment which we presume referring to the 77% male subjects. We have included the following in the revised limitation section: 

Page 23; lines 375-377: 

“Finally, the study population included 77% males and although we did not find any significant effect modification by sex, there could be variations in the TMAO pathway with sex, which might have impacted our results.”

---

## [Decision Letter · Decision Letter 1]

9 Dec 2020

Plasma trimethylamine N-oxide and its metabolic precursors and risk of mortality, cardiovascular and renal disease in individuals with type 2-diabetes and albuminuria

PONE-D-20-21640R1

Dear Dr. Winther,

We’re pleased to inform you that your manuscript has been judged scientifically suitable for publication and will be formally accepted for publication once it meets all outstanding technical requirements.

Kind regards,

M. Faadiel Essop

Academic Editor

PLOS ONE

Additional Editor Comments (optional):

Reviewers' comments:

Reviewer's Responses to Questions

**Comments to the Author**

1. If the authors have adequately addressed your comments raised in a previous round of review and you feel that this manuscript is now acceptable for publication, you may indicate that here to bypass the “Comments to the Author” section, enter your conflict of interest statement in the “Confidential to Editor” section, and submit your "Accept" recommendation.

Reviewer #3: All comments have been addressed

2. Is the manuscript technically sound, and do the data support the conclusions?

Reviewer #3: Yes

3. Has the statistical analysis been performed appropriately and rigorously? 

Reviewer #3: Yes

4. Have the authors made all data underlying the findings in their manuscript fully available?

Reviewer #3: (No Response)

5. Is the manuscript presented in an intelligible fashion and written in standard English?

Reviewer #3: Yes

6. Review Comments to the Author

Reviewer #3: (No Response)

7. PLOS authors have the option to publish the peer review history of their article (what does this mean?). If published, this will include your full peer review and any attached files.

Reviewer #3: No

---

## [Editor Report · Acceptance letter]

22 Feb 2021

PONE-D-20-21640R1 

Plasma trimethylamine N-oxide and its metabolic precursors and risk of mortality, cardiovascular and renal disease in individuals with type 2-diabetes and albuminuria 

Dear Dr. Winther:

I'm pleased to inform you that your manuscript has been deemed suitable for publication in PLOS ONE. Congratulations! Your manuscript is now with our production department. 

Kind regards, 

on behalf of

Dr. M. Faadiel Essop 

Academic Editor

PLOS ONE